# Subchronic Oral Cylindrospermopsin Exposure Alters the Host Gut Microbiome and Is Associated with Progressive Hepatic Inflammation, Stellate Cell Activation, and Mild Fibrosis in a Preclinical Study

**DOI:** 10.3390/toxins14120835

**Published:** 2022-12-01

**Authors:** Punnag Saha, Macayla Upright, Dipro Bose, Subhajit Roy, Ayushi Trivedi, Madhura More, Geoff I. Scott, Bryan W. Brooks, Saurabh Chatterjee

**Affiliations:** 1Environmental Health and Disease Laboratory, Department of Environmental and Occupational Health, Program in Public Health, Susan and Henry Samueli College of Health Sciences, University of California, Irvine, CA 92697, USA; 2Toxicology Core, NIEHS Center for Oceans and Human Health on Climate Change Interactions, Department of Environmental and Occupational Health, Program in Public Health, Susan and Henry Samueli College of Health Sciences, University of California, Irvine, CA 92697, USA; 3Department of Environmental Health Sciences, Arnold School of Public Health, University of South Carolina, Columbia, SC 29208, USA; 4Department of Environmental Science, Baylor University, Waco, TX 76798, USA

**Keywords:** cylindrospermopsin, gut microbiome, HMGB1, hepatotoxicity

## Abstract

Epidemiological studies have reported a strong association between liver injury and incidences of hepatocellular carcinoma in sections of humans globally. Several preclinical studies have shown a strong link between cyanotoxin exposure and the development of nonalcoholic steatohepatitis, a precursor of hepatocellular carcinoma. Among the emerging threats from cyanotoxins, new evidence shows cylindrospermopsin release in freshwater lakes. A known hepatotoxin in higher concentrations, we examined the possible role of cylindrospermopsin in causing host gut dysbiosis and its association with liver pathology in a mouse model of toxico-pharmacokinetics and hepatic pathology. The results showed that oral exposure to cylindrospermopsin caused decreased diversity of gut bacteria phyla accompanied by an increased abundance of *Clostridioides difficile* and decreased abundance of probiotic flora such as *Roseburia*, *Akkermanssia,* and *Bacteroides thetaiotamicron*, a signature most often associated with intestinal and hepatic pathology and underlying gastrointestinal disease. The altered gut dysbiosis was also associated with increased Claudin2 protein in the intestinal lumen, a marker of gut leaching and endotoxemia. The study of liver pathology showed marked liver inflammation, the release of damage-associated molecular patterns, and activation of toll-like receptors, a hallmark of consistent and progressive liver damage. Hepatic pathology was also linked to increased Kupffer cell activation and stellate cell activation, markers of progressive liver damage often linked to the development of liver fibrosis and carcinoma. In conclusion, the present study provides additional evidence of cylindrospermopsin-linked progressive liver pathology that may be very well-linked to gut dysbiosis, though definitive evidence involving this link needs to be studied further.

## 1. Introduction

Cyanobacteria, commonly known as ‘blue-green algae’ are primarily prokaryotic, photoautotrophic organisms found in all sorts of water systems [1]. Climate change-related stressors in combination with global warming, anthropological activities, and eutrophication (accumulation of excess nutrients) potentiate the excessive growth of these cyanobacteria present in the water bodies, a phenomenon also known as the formation of harmful algal blooms (HAB) [2,3]. HAB formation exerts various deleterious effects on aquatic systems. First, HABs create a hypoxic environment in the water body as a consequence of excessive utilization of dissolved oxygen in the water to facilitate their uncontrolled growth, leading to the death of aquatic organisms [4]. Most importantly, HABs also produce a wide array of cyanotoxins in the form of secondary metabolites, which are a major concern for public health officials as these toxins can severely affect the ecosystem and human health across the world [5,6]. Among the prominent cyanotoxins, microcystin, nodularin, and cylindrospermopsin (CYN) are well-known hepatotoxins, whereas saxitoxin and anatoxin-a are categorized as neurotoxins [7].

CYN was first isolated from the cyanobacterium *Cylindrospermopsis raciborskii* in 1992. Since then, various other species of cyanobacteria including *Umezakia natans*, *Raphidiopsis curvata, Aphanizomenon ovalisporum*, *Anabaena lapponica*, *Aphanizomenon flos-aquae,* etc., also have been reported as CYN-producing organisms [8,9,10]. Structure-wise, CYN is an alkaloid that primarily includes a positively charged tricyclic guanidine moiety linked to the hydroxymethyluracil group and a negatively charged sulfate moiety with a molecular weight of 415 kDa. Thus, CYN can exist in zwitterionic form, which explains its highly water-soluble nature. In a study conducted by Chiswell et al., CYN was found to be unaffected by changes in pH and remained in relatively stable conditions in varying temperatures from 4 °C to 50 °C for up to five weeks. In addition, boiling the CYN-contaminated water for 15 min was also found to be ineffective to degrade the toxin. The presence of CYN has been detected in the surface water samples from all continents worldwide including Asia, Africa, Europe, North America, South America, Oceania, and even Antarctica [11,12,13,14,15,16,17]. In a study conducted by Loftin et al., 1161 lakes were surveyed nationwide across the United States to detect the presence of cyanotoxins in the water bodies and CYN was detected in 4% of the samples, with a mean concentration of 0.56 µg/L, whereas potential CYN-producing cyanobacteria were found in 67% of the samples [18].

The possible routes of CYN exposure in humans are chronic exposure to the toxin via consumption of finished drinking water, accidental inhalation, ingestion or dermal contact with the toxin during recreational activities (e.g., swimming, boating), consumption of contaminated foods, vegetables, or aquatic organisms (e.g., fish, shellfish) [19]. However, consumption of contaminated drinking water is often considered the major route of cyanotoxin exposure, which may result in various deleterious effects in humans including fever, headache, vomiting, bloody diarrhea, damage to vital organs such as the liver and kidney, and electrolyte imbalance.

Although primarily known to be a hepatotoxin only in the initial periods, recent studies uncovered a wide array of CYN-associated toxicities which encompass pro-genotoxicity [20], cytotoxicity [21], neurotoxicity [22], and immunotoxicity [23]. Mechanistically, CYN exerts its cytotoxic effects by inhibiting protein synthesis [24] confirmed by both in vitro and in vivo studies, as well as inhibiting the antioxidant glutathione synthesis [25,26] and reactive oxygen species generation (ROS) [27,28,29] as indicated by many in vitro studies. CYN was also found to cause micronuclei formation in the human lymphoblastoid cell line [20], human liver-derived HepaRG cells, and colon-derived Caco-2 cells [30], proving its pro-genotoxic effects. In a study conducted by Tasker et al., the authors showed that a low dose of CYN was able to cause a pro-inflammatory response and induce apoptosis alone or in combination with other cyanotoxins in the BV-2 murine microglia cells and Neuro2a murine neuroblasts cells [31], indicating its immunotoxic and potential neurotoxic nature.

The gut microbiome, currently referred to as the ‘second brain’ [32], has earned a lot of attention in recent times due to its vast importance in various disease pathophysiologies [33,34]. The gut microbiome consists of different bacterial, viral, fungal, and protozoal species residing where the bacterial population (bacteriome) is the major microbial population. Interaction between the host and the gut microbiome is highly important for the host as the microbiome potentiates numerous beneficial aspects for the host including the synthesis of vitamins and short-chain fatty acids, nutrient absorption, and immune function. Although the gut microbial pattern changes with age, a sudden ‘dysbiosis’ or alteration in the gut microbiome composition due to chronic exposure to environmental factors such as cyanotoxins can influence the host’s health negatively [35,36]. Interestingly, dysbiosis-mediated alterations in the host are not only limited to the intestinal microenvironment only but also are linked to various other organs including the liver and the brain. The ‘gut–liver axis’ is the bidirectional mode of crosstalk between the intestine and the liver where the gut-derived molecules are carried to the liver via the portal vein and the liver responds via the biliary systems to the gut [37]. Although many studies have been conducted to establish potential mechanisms of CYN-mediated toxicity since the first CYN-related outbreak in Palm Island, Australia in 1979 [38], no studies have been conducted so far on the outcomes of subchronic exposure to CYN on the gut microbiome.

Therefore, in this current study, we sought to determine whether subchronic CYN administration in mice for a continuous 15 days was able to affect the host’s gut microbiome signature and if the resulting dysbiosis was able to further aid in the CYN-mediated hepatotoxicity in mice in a synergistic manner via the gut–liver axis.

## 2. Results

### 2.1. CYN Administration in Mice Resulted in an Altered Gut Microbiome Signature

For our study, we wanted to determine whether oral administration in mice for a consecutive period of 15 days contributed to the alteration of the host gut microbiome profile. Next-gen sequencing was performed using the fecal pellets obtained from mice for both experimental groups. At the phylum level, we observed an increased relative abundance of the phyla Firmicutes and Actinobacteria with a parallel decreased abundance of phyla Bacteroidetes and Verrucomicrobia in the LEAN+CYN group compared to the LEAN control group (Figure 1A). At the family level analysis, a decreased abundance of Muribaculaceae, Bacteroidaceae, and Akkermansiaceae was observed, whereas the abundance of Lactobacillaceae and Bifidobacteriaceae families was found to be increased in the LEAN+CYN group of mice compared to the LEAN mice (Figure 1B).

Next, we sought to estimate the overall changes in the gut microbiome composition between the two experimental groups of mice. α-Diversity, represented by the Shannon diversity index, was found to be more significantly decreased in the LEAN+CYN group of mice than in the control LEAN group (Figure 1C, * *p* < 0.05). However, no significant difference was observed in the case of β-diversity (Figure 1D).

Then, we wanted to observe the effect of CYN exposure at the species level in the experimental groups of mice. The relative abundance of individual species was calculated for both groups and represented as the percentage relative abundance for each group. *Akkermasia muciniphilia*, which is known for its role in intestinal mucin layer protection, gut barrier integrity maintenance, and regulation of the host’s gut immune response [39,40,41], was found to be significantly decreased in the percentage relative abundance in the LEAN+CYN mice compared to the LEAN mice (Figure 2A, *** *p* < 0.001). *Bacteroides thetaiotaomicron*, another gut commensal known to be associated with utilizing polysaccharides to simpler forms of carbohydrates, thereby contributing to the host’s carbohydrate metabolism [42], was also found markedly decreased in the percentage relative abundance in the LEAN+CYN group compared to the LEAN group (Figure 2B, *** *p* < 0.001). *Blautia coccoides* are commonly found in the murine intestinal tract and can aid in the host’s lipid metabolism [43]. In our study, the percentage relative abundance of *B. coccoides* was significantly lowered in the CYN-exposed mice than in the vehicle-treated control LEAN mice (Figure 2C, *** *p* < 0.001). Various gut commensal bacteria including *Schaedlerella arabinosiphila*, *Duncaniella muris*, *Enterococcus faecium*, and *Muribaculaceae_u_s* were also found to be markedly decreased in percentage relative abundance in the LEAN+CYN group of mice compared to the LEAN group (Figure 2D–G, *** *p* < 0.001). In addition, we also observed a significantly decreased percentage relative abundance of the probiotic bacteria *Oscillibacter_u_s* and *Parabacteroides_u_s* in the LEAN+CYN mice compared to the LEAN group of mice (Figure 2H,I, *** *p* < 0.001). Interestingly, the percentage relative abundance of the butyrogenic bacterium *Roseburia_u_s* was found to be markedly decreased with a parallel increase in abundance of the opportunistic pathogen *Clostridioides difficile* in the CYN-exposed mice (Figure 2J,K, *** *p* < 0.001).

### 2.2. Exposure to the Environmental Toxin CYN Caused Gut Leaching with a Concomitant Increased Systemic Release of Pathogen-Associated Molecular Pattern (PAMP) and Damage-Associated Molecular Pattern (DAMP) in Mice

The epithelial lining of the small intestine, composed of neighboring intestinal epithelial cells (IECs), acts as the primary absorption site of nutrients and as a physical barrier against invading pathogens and extracellular substances [44]. Neighboring IECs are joined by tight junction (TJ) proteins and any alteration in these TJ proteins leads to a ‘leaky gut’ phenotype [45]. Our result showed a marked alteration in the epithelial TJ protein expression where increased expression of Claudin 2 (Figure 3A,B, *** *p* < 0.001) with a parallel decrease in Occludin expression (Figure 3C,D, *** *p* < 0.001) was observed by immunofluorescence method in the LEAN+CYN group compared to the LEAN group. In addition, significantly increased expression of the DAMP high mobility group box 1 (HMGB1) in the intestinal lumen was also detected for the CYN-exposed mice compared to the LEAN mice (Figure 3E,F, *** *p* < 0.001) by immunohistochemistry. An altered gut microbiome pattern resulting from dysbiosis with a leaky gut often leads to gut leaching and systemic rise of PAMPs, e.g., endotoxins and DAMPs [46,47,48]. Following this rationale, we wanted to find out whether a CYN-associated change in gut microbiome profile led to increased endotoxemia and DAMP release in circulation. Indeed, we detected a markedly increased endotoxemia in the LEAN+CYN group of mice compared to the LEAN control mice (Figure 3G, * *p* < 0.05) by performing a LAL assay. The circulatory level of the DAMP HMGB1 in the CYN-exposed mice was found to be significantly elevated compared to the vehicle-treated LEAN mice, as detected by ELISA using the serum samples (Figure 3H, *** *p* < 0.001). In addition, the Shannon diversity of the gut microbiome negatively correlated with both endotoxin and HMGB1 levels in serum (r = −0.7799, −0.762 respectively), suggesting a less-diverse gut microbiome profile resulting from CYN administration was associated with the increased endotoxemia and HMGB1 concentration in these mice (Figure 3I,J).

### 2.3. CYN Administration in Mice Exhibited Markers of Liver Injury, Mild Fibrosis, and Caused Kupffer Cell and Stellate Cell Activation

Following gut microbiome analysis and detection of elevated levels of endotoxin and HMGB1 in serum, we wanted to observe the pathophysiology exerted by the environmental hepatotoxin CYN. Liver histopathology was assessed by performing hematoxylin and eosin (H&E) staining, where the CYN-exposed group of mice showed increased rarification of cytoplasm along with vacuolation compared to the LEAN group, indicating signs of liver damage (Figure 4A). Mild fibrosis in the portal region of the liver also was detected by Picrosirius red staining in the mice of the LEAN+CYN group, which was more significantly increased than in the LEAN mice group, where no collagen deposition was observed (Figure 4B,E, *** *p* < 0.001). Further, we investigated whether oral administration of CYN in mice resulted in the activation of both Kupffer cells (KC) and hepatic stellate cells (HSC), which are regarded as the hallmarks of pathophysiological outcomes in the liver. Expression of both clusters of differentiation 68 (CD68) (KC activation marker) and α-smooth muscle actin (α-SMA) (HSC activation marker) in the liver sections of the LEAN+CYN group was found to be more markedly heightened than in the LEAN group, as obtained by performing the immunohistochemistry method (Figure 4C,D,F,H, *** *p* < 0.001). These results were further confirmed by doing qRT-PCR analysis, where gene expression of both CD68 and α-SMA was again found to be significantly increased in the LEAN+CYN mice compared to the LEAN control mice (Figure 4G,I, * *p* < 0.05, *** *p* < 0.001). Shannon diversity of the gut microbiome also was correlated negatively with both CD68 and α-SMA immunoreactivities (r = −0.619, −0.5808 respectively), indicating CYN-mediated decreased α-diversity in the gut has a strong association with increased KC and HSC activation in the liver (Figure 4J,K).

### 2.4. CYN-Induced Increase of Circulatory PAMP and DAMP Levels Led to Hepatic Inflammasome Activation, Thus Linking Hepatic Inflammation

Gut-derived endotoxin, transported to the liver by the portal vein, can act as the initial signal for the initiation of hepatic inflammasome activation by increasing the expression of NLR family pyrin domain containing 3 (NLRP3). A second signal, mediated by DAMPs, further triggers the hepatic inflammation activation event [49]. Following this rationale, we sought to determine whether the increased systemic levels of PAMP (endotoxin) and DAMP (HMGB1), as observed in our study, triggered the inflammasome activation in the CYN-exposed group of mice. Dual labeling was performed by using the immunofluorescence staining technique and the results showed a significantly increased co-localization event of NLRP3 (red) and ASC2 (green), represented by yellow dots, in the liver slices of the LEAN+CYN mice compared to the control LEAN group of mice (Figure 5A,B, *** *p* < 0.001) indicating clear inflammasome activation. Activation of inflammasome results in the release of the pro-inflammatory cytokine Interleukin-1β (IL-1β), which then acts in both autocrine and paracrine manner, contributing to inflammatory surge and HSC activation in the liver [50]. As detected by the immunohistochemistry method, a significantly heightened expression of interleukin-1β (IL-1β) was observed in the liver sections of the CYN-exposed group compared to the LEAN control group (Figure 5C,D, *** *p* < 0.001). This result was then further confirmed by gene expression studies using the qRT-PCR method. mRNA levels of both IL-1β and interleukin-6 (IL-6) pro-inflammatory cytokines were found to be more significantly increased in the livers of the LEAN+CYN group of mice than those of the LEAN mice (Figure 5E, *** *p* < 0.001). In addition, the Shannon diversity depicting gut microbiome composition negatively correlated with both NLRP3-ASC2 co-localization events and IL-1β immunoreactivity (r = −0.7511, −0.6519 respectively), implying that the CYN-associated microbiome alteration may be aiding in increased inflammasome activation and resulting inflammation in the liver (Figure 5F,G).

### 2.5. CYN Exposure in Mice Increased TGF-β Expression in the Liver, Leading to Smad2/3-Smad4 Mediated Fibrotic Pathway Activation

The pleiotropic cytokine transforming growth factor-beta (TGF-β) is induced by persistent liver injury and plays a central role in the activation of HSC as a wound healing process, but ultimately leads to liver fibrosis [51]. Our results revealed a significantly increased expression of TGF-β in the liver slices of the LEAN+CYN group of mice in both protein level and mRNA level compared to the LEAN group of mice (Figure 6C,D,E, *** *p* < 0.001), as detected by the immunohistochemistry and qRT-PCR methods, respectively. Further, TGF-β activates the downstream Smad2/3-Smad4-mediated fibrotic pathway marked by the deposition of collagen in the liver [52]. Dual labeling of Smad2/3 (red) and Smad4 (green) by immunofluorescence staining method in the liver sections exhibited markedly increased Smad2/3-Smad4 co-localization events (marked by yellow dots) in the CYN-exposed group of mice compared to the vehicle-treated LEAN control mice (Figure 6A,B, * *p* < 0.05). A negative correlation was estimated between the Shannon diversity and both TGF-β immunoreactivity, and Smad2/3-Smad4 co-localization, (r = −0.6084, −0.6995 respectively), suggesting an association between activation of the fibrotic pathway and decreased α-diversity of the gut microflora as a result of CYN treatment in mice (Figure 6F,G).

### 2.6. CYN Treatment in Mice Triggered the Intrinsic Apoptotic Pathway in the Liver

An inflamed microenvironment in the liver, mediated by persistent and increased expression of pro-inflammatory cytokines by KC and other immune cells, often contributes to the apoptosis of hepatocytes [53]. In our study, we wanted to observe whether CYN administration in mice and resulting hepatic inflammation augmented the intrinsic apoptosis signaling pathway. Western blot results showed that pro-apoptotic protein expression (Bax:Bcl-2) was significantly increased in the LEAN+CYN group than that of the LEAN control group (Figure 7A,B, *** *p* < 0.001). Next, protein expressions of both initiator Caspase of the intrinsic apoptosis pathway (Cleaved-Caspase 9 expression normalized against Total-Caspase 9 expression) and executioner Caspase (Cleaved-Caspase 3 expression normalized against Total-Caspase 3) were found to be markedly elevated in the CYN-exposed mice group compared to the vehicle-treated LEAN control group (Figure 7A,D,E, *** *p* < 0.001). Finally, Caspase-mediated poly adenosine diphosphate-ribose polymerase (PARP) cleavage (Cleaved-PARP expression normalized against Total-PARP), a marker of the late phase of apoptosis was also found to be significantly increased in the LEAN+CYN group of mice compared to the LEAN control group of mice (Figure 7A,C, *** *p* < 0.001), thereby confirming the CYN-mediated activation of the intrinsic apoptosis pathway in the liver.

### 2.7. CYN Administration in Mice Caused Elevated Extracellular HMGB1 Expression and HMGB1-Mediated Receptor Expression in the Liver

Extracellular secretion of the alarmin HMGB1 may occur actively by the innate immune cells or passively by apoptotic cells, which leads to the binding of the secreted HMGB1 to its receptors and initiate multiple inflammatory signaling pathways [54,55]. In our study, we wanted to detect the expression of HMGB1 in the mice liver sections as increased HMGB1 level was earlier detected in both the small intestine and circulation. Indeed, HMGB1 expression was found to be markedly elevated in the LEAN+CYN mice compared to the LEAN group as detected by both immunohistochemistry (Figure 8A,B, *** *p* < 0.001) and Western blot methods (Figure 8C,D, *** *p* < 0.001). Extracellular HMGB1 can bind to both receptors for advanced glycation end products (RAGE) and toll-like receptor 4 (TLR4) receptors, which in turn interact with the common adapter molecule myeloid differentiation primary response protein (MyD88) for the downstream signaling process. Our results indicated a significantly increased protein expression of both TLR4 (Figure 8C,E, * *p* < 0.05) and RAGE receptors (Figure 8C,F, *** *p* < 0.001) in the LEAN+CYN mice compared to the LEAN group. Next, protein expression of the adapter molecule MyD88 was also detected to be significantly elevated in the LEAN+CYN group of mice compared to the LEAN mice (Figure 8C,G, * *p* < 0.05) by Western blot. These results imply the role of increased extracellular HMGB1 as a result of CYN administration in mice and its possible participation in CYN-mediated hepatotoxicity.

## 3. Discussion

The intestinal microbiome encompasses a diverse variety of microorganisms, including bacteria, archaea, fungi, and protozoa, colonizing the small and large intestines. The intestine harbors more than 100 trillion commensal bacteria, which form the major population among the other microorganisms and play a pivotal role in the host’s metabolism, immunity, physiology, and nutrition [56]. Several reports have linked the alteration in the gut microbiome (also known as ‘gut dysbiosis’) to metabolic conditions such as obesity [57], and a wide range of diseases including non-alcoholic fatty liver disease (NAFLD) [58], irritable bowel syndrome (IBD) [59], inflammatory bowel disease (IBS), and [60] neuropathological conditions such as Alzheimer’s disease [61] and Parkinson’s disease [62], proving the immense effects of the intestinal microflora. However, apart from pathological conditions, environmental exposure also greatly influences the host microbiome. Over the past few years, our group has extensively shown that environmental factors such as exposure to the cyanotoxin microcystin-LR (MC-LR) [35,36] and pesticides such as permethrin [63,64,65,66,67] also cause marked alteration in the host’s gut microbiome composition, resulting in numerous adverse effects on the host’s overall pathophysiology.

In this study, we report for the first time a significant effect of the cyanotoxin CYN on the intestinal microflora as a result of a subchronic exposure via the oral route, a viable route of exposure based on existing epidemiological studies. Our results clearly indicated a plausible gut dysbiosis scenario in the intestinal microenvironment where we observed a marked increase in the relative abundance of the phyla Firmicutes and Actinobacteria with a parallel decrease in Bacteroidetes and Verrucomicrobia abundance. The majority of the gut commensal bacteria belonged to the Firmicutes and Bacteroidetes phyla [68] and any increase or decrease in the Firmicutes/Bacteroidetes (F/B) ratio in abundance is often regarded as a key indicator of the host’s microbiome signature [69]. A decreased F/B ratio that represents increased Firmicutes abundance over that of Bacteroidetes has been reported in obese individuals, IBD patients, and has been found in our current study too. Loss of microbiota diversity has been reported to be associated with several metabolic diseases, including IBD [70], IBS [71], celiac disease [72], type 1 diabetes mellitus [73], obesity [74], etc. Interestingly, α-diversity marked by the Shannon diversity index was found significantly decreased in the CYN-exposed mice in the current study, showing a similar pattern with the previously mentioned diseased conditions. However, we did not observe any significant alteration in the case of β-diversity.

In the species-level analysis, several salient intestinal symbionts including *A. muciniphilia*, *B. thetaiotamicron*, *B. coccides*, *Oscillibacter_u_s*, *Parabacteroidetes_u_s*, *Roseburia_u_s*, and murine gut commensals including *S. arabinosiphilia*, *D. muris*, *E. faecium*, and *Muribaculaceae_u_s*, were found to be significantly decreased in abundance in the CYN-exposed mice, which explicitly indicate a clear scenario of altered gut microbiome profile. *A. muciniphilia*, a gut commensal belonging to the phylum Verrucomicrobia, degrades the mucin layer of the epithelial lining in the intestines into simpler metabolites, which in turn serve as the energy source for other gut commensals and promote increased mucin production by intestinal Goblet cells to protect the epithelial lining [75]. Increased abundance of this bacterium is known to be associated with increased host immunity, gut barrier improvement, and cellular lipid metabolism [41]. Thus, a decrease in the abundance of *A. muciniuphilia* due to CYN treatment might result in increased gut permeability and thinning of the protective mucin layer in the gut. Gut commensals such as *B. thetaiotamicron* and *B.coccides* explicitly aid in the host’s metabolism. *B. thetaiotamicron* is known to contain various glycoside hydrolases and polysaccharide lyases, thereby digesting the dietary complex carbohydrates into simpler forms that can be used by other gut commensals as metabolites [76]. In a study conducted by Durant et al., the authors established a cross-kingdom interaction between the host intestinal epithelial cells and *B. thetaiotamicron* via bacterial outer membrane vesicles (OMVs) produced by the bacterium that helped in anti-inflammatory cytokine IL-10 production, thereby maintaining a “healthy” status in the control group but found to be absent in the IBD patients [77]. Thus, a CYN-mediated decrease in the abundance of this bacterium might hamper the host’s metabolism and gut immune health. However, *B. thetaiotamicron* is also known to be an opportunistic pathogen [78] and found to be harboring multiple anti-microbial resistance genes as a result of cyanotoxin (MC-LR) treatment, as reported in a recent study conducted by our group [36]. Thus, more in-depth studies are needed to uncover the exact role of this bacterium in the intestinal microenvironment because of cyanotoxin exposure. Murine intestinal bacterium *B. coccides* was reported to metabolize glycosylceramides successively to ceramides, simpler fatty acids, and sphingoid bases, which contributed to improved intestinal health in a study conducted by Hamajima et al. [43]. Probiotic bacteria such as *Oscillibacter* and *Parabcteroides* are associated with enhancing the activity of IL-10-producing Treg cells, thus maintaining an anti-inflammatory condition in the intestinal microenvironment [79]. *Roseburia* is a known producer of the short-chain fatty acid (SCFA) butyrate, which acts as the primary energy source for the host’s colonocytes and exerts anti-inflammatory properties via NF-κB inhibition through G-protein-coupled receptor signaling in the intestine [80]. Interestingly, in our present study, we also detected an increased abundance of the opportunistic pathogen *C. difficile* in the CYN-treated mice. The bacterium *C. difficile* is able to inhabit naturally the intestines of neonates. However, the *C. difficile* population starts to decline with increased age as other gut commensals can outgrow the bacterium competitively as they colonize in the intestines [81,82]. Any disturbance in the normal microbiota signature can lead to decreased colonization resistance and potentiate a suitable environment for *C. difficile*-mediated infection in the host if the bacterium is already present in the gut. Hence, decreased abundance of the beneficial bacterial populations with a simultaneous increase in opportunistic pathogen abundance in the gut due to oral CYN exposure can be postulated to create an imbalance in the normal physiological functions, disrupting intestinal homeostasis, and as likely to drive the intestinal microenvironment to a greater risk of pathological outcomes. Importantly, the association between the gut microbial α-diversity and liver immune markers depicted a noticeably strong negative correlation, suggesting a decreased diverse gut microbiome is associated with increased liver pathophysiological conditions in mice exposed to CYN. These correlation analyses in combination with the species abundance data suggest that the decreased α-diversity of the gut microbiome due to subchronic CYN exposure might have been caused as a result of selection pressure on the gut microbiome by the cyanotoxin that promoted the abundance of pathobionts such as *C. difficile* with a parallel depletion in the beneficial species that affected the liver pathophysiology. However, to establish the mechanistic link between the direct role of the gut microbiome and how dysbiosis can be associated with organ-specific toxicity, use of the gnotobiotic mice models for CYN-associated toxicity could prove very helpful in the future.

We and other research groups have shown that dysbiosis of normal gut microflora due to environmental exposure, pathological conditions, or antibiotic treatment can lead to an altered expression of the epithelial TJ proteins causing a “leaky gut” phenotype [35,36,67,83]. In this current study, we detected an increased expression of the TJ protein Claudin2 with a concomitant decrease in Occludin level in the intestinal epithelial lining of the CYN-administered mice that aligned with the previously reported findings confirming the leaky gut scenario. However, the probable cause of the gut leakiness in this study was difficult to point out as it is still unknown whether CYN alone could directly affect the TJ protein expression in IEC, or whether the CYN-mediated gut microbiota alteration resulted in the gut leaching phenomenon in the mice as an indirect effect of the toxin exposure, although a combination of both mechanisms cannot be ruled out at this point of study. These hypotheses can be further investigated through in-depth mechanistic studies, which will help toxicologists to comprehend better the overall toxicity of CYN in the future. We also detected a significantly increased secretion of the DAMP HMGB1 in the intestinal lumen of CYN-exposed mice compared to the control mice, indicating a plausible hyper-inflamed state in the small intestine. Furthermore, markedly increased levels of endotoxemia and serum HMGB1 levels were noted as a consequence of CYN toxicity in mice, which confirms the increased gut-leaching effect in this study.

Pronounced CYN-mediated hepatotoxic effects were also observed in the CYN-treated group of mice. First, distinct signs of liver injury in the treated mice were indicated by the histopathological results. Then, we confirmed the heightened hepatic damage exerted by CYN by detecting the markedly increased expression level of the intrinsic apoptosis pathway markers, including BAX:Bcl-2, Cleaved PARP, Cleaved Caspase 9, and Cleaved Caspase 3. In addition, we also observed an increased immunoexpression of the DAMP HMGB1 and its canonical receptors RAGE and TLR4 in the livers of the CYN-administered mice as a result of the intrinsic apoptotic cell death, further corroborating a sustained scenario of hepatic injury as a result of subchronic exposure to CYN for 15 days. Interestingly, elevated levels of HMGB1 were previously found in the serum samples and respective small intestines of CYN mice, as mentioned earlier. Overall, these results clearly indicated that the alarmin HMGB1 can possibly act both in a paracrine manner, being a possible mediator of the gut-liver axis, as well as in an autocrine manner by its release from the apoptotic dead cells. However, tracing the exact origin of HMGB1 could not be performed at this point, which can be regarded as a limitation of the present study. Kupffer cells, the resident macrophages of the liver, were found to be in the “activated” or pro-inflammatory stage, as marked by significantly increased CD68 expression on both mRNA and protein levels. This possibly occurred as the direct consequence of both CYN-associated liver injury and the increased presence of gut-derived endotoxin and HMGB1 circulated to the liver via the portal vein. We also detected increased expression of the pattern recognition receptor (PRR) TLR4 protein, which precisely identifies endotoxins, with a parallel increased expression of its adapter MyD88 in the liver lysates, which further proved the activation of the TLR4-Myd88 pathway and possible activation of the downstream NF-κβ pathway (not shown in this study) as a result of CYN-mediated toxicity in mice. The NLRP3 inflammasome can be activated by a two-signal model where the first signal is initiated by the PAMPs through PRRs and the second signal is then triggered by DAMPs as one of the possible mediators [84,85]. Interestingly, we also observed increased co-localization of the NLRP3 inflammasome with its adapter ASC2, proving that activation of the NLRP3 inflammasome complex occurred in the LEAN+CYN mice possibly due to the presence of elevated endotoxin and HMGB1 levels in that group. Consequently, the activation of the NLRP3 inflammasome complex led to a heightened inflammatory response in the livers of toxin-exposed mice, as elevated levels of the pro-inflammatory cytokine IL-1β and IL-6 were detected. Sustained liver injury in combination with inflammation often leads to activation of the HSCs [86]. The above effect was also observed in the present study as increased expression of α-SMA was found in the liver sections of the CYN-exposed mice. An increased TGF-β expression and subsequently elevated co-localization of Smad2/3 and Smad4 proteins in the livers of CYN mice also implied the possible role of the TGF-β mediated activation of the Smad fibrotic pathway. Indeed, early onset of fibrosis was noted in the portal region of the livers of CYN-treated mice by Picrosirius red Staining, suggesting the possible role of CYN in play.

In conclusion, this study confirmed a novel aspect of cyanotoxin CYN exposure by showing its fascinating role in intestinal gut microbiome alteration, a phenomenon that acts parallelly with the normal hepatoxicity mediated by the toxin and further contributes greatly to the overall toxicity (Figure 9). The study also strengthens the increasingly acclaimed notion among the clinical community that cyanotoxins are associated with liver disease and can lead to an aggressive progression to cirrhosis and hepatocellular carcinoma.

## 4. Materials and Methods

Materials: CYN was obtained from Enzo Life Sciences (Farmingdale, NY, USA). Anti-Claudin 2, anti-Occludin, anti-HMGB1, anti-CD68, anti-α-SMA, and anti-NLRP3 primary antibodies were purchased from Abcam (Cambridge, MA, USA). Anti-IL-1β, anti-ASC2, anti-Smad2/3, anti-Smad 4, anti-TGF-β, and anti-TLR4 primary antibodies were bought from Santacruz Biotechnology (Dallas, TX, USA) whereas anti-β-actin and anti-Bcl-2 antibodies were purchased from Proteintech (Rosemont, IL, USA). Anti-BAX, anti-Cleaved PARP, anti-PARP, anti-Cleaved Caspase 9, anti-Caspase 9, and anti-Cleaved Caspase 3, anti-Caspase 3 antibodies were obtained from Cell Signaling Technology (Danvers, MA, USA). Anti-MyD88 primary antibody was purchased from Abclonal Technology (Woburn, MA, USA) whereas anti-RAGE was obtained from Thermo Fisher Scientific (Waltham, MA, USA). Species-specific biotinylated conjugated secondary antibody and Streptavidin-horseradish peroxidase (Strp-HRP) (Vectastain Elite ABC kit) were purchased from Vector laboratories (Burlingame, CA, USA). Fluorescence conjugated Alexa Fluor secondary antibodies, and ProLong Diamond antifade mounting media with 4′,6-diamidino-2-phenylindole (DAPI), were bought from Thermo Fisher Scientific (Waltham, MA, USA). If not specified otherwise, all the necessary chemicals used for this study were purchased from Sigma-Aldrich (St. Louis, MO, USA). Murine liver tissues were processed at AML Laboratories (St. Augustine, FL, USA) for paraffin-embedding and sectioning into slides. Picrosirius red staining was carried out at the Instrument Resources Facility, University of South Carolina School of Medicine (Columbia, SC, USA). Bacteriome analysis was performed by Cosmos ID (Rockville, MD, USA).

### 4.1. Animals

Pathogen-free, adult (12 weeks old), wild-type (WT), male C57BL/6J mice were purchased from Jackson Laboratories (Bar Harbor, ME, USA) for conducting this study. All the mice experiments for this study were performed strictly following the National Institutes of Health (NIH) guidelines for humane care and use of laboratory animals and local Institutional Animal Care and Use Committee (IACUC) standards (Animal Protocol Number: 2488-101501-051220; approval date: 5 October 2021). The animal handling procedures for this study were approved by the University of South Carolina (Columbia, SC, USA). After arrival at the facility, all mice were housed inside 3 mice/cages in a 22–24 °C temperature-controlled room with a 12 h light/12 h dark cycle and had ad libitum access to food and water throughout the course of the study. Every mouse receiving treatment was sacrificed once the dosing was completed for this study. Following anesthesia, blood was collected from individual mice using the cardiac puncture method, and serum samples were isolated from the freshly obtained blood. All serum samples were kept at −80 °C for further analysis. Liver and small intestine samples from individual mice were collected post-euthanization and then fixed in 10% neutral buffered formaldehyde (Sigma-Aldrich, St. Louis, MO, USA). Fecal pellets were also collected from the colon of each mouse and preserved at −80 °C for microbiome analysis.

### 4.2. Experimental Murine Model of CYN Exposure

After one week period of acclimatization post-arrival, 12 mice were randomly divided into two groups containing 6 mice per group (*n* = 6). The control group of mice (LEAN) received only vehicle (phosphate-buffered saline (PBS)) whereas the mice of the treated group (LEAN+CYN) were administered with CYN (60 µg/kg body weight; diluted in PBS) for a continuous 15-day period by oral gavage route. In a study conducted by Humpage et al. using male Swiss albino mice, the no observed adverse effect level (NOAEL) for CYN-associated toxicity was found to be 30 μg/kg/day whereas the lowest observed adverse effect level (LOAEL) was 60 μg/kg/day in terms of changes in kidney weight [87]. Hence, the concentration of 60 μg/kg body weight/day was selected as the most favorable dose for our murine model of CYN exposure. For this study, each mouse was dosed with 100 µL of the vehicle or CYN according to their preassigned experimental group and fed with only a chow diet throughout the course of the study.

### 4.3. Bacteriome Analysis

As described in one of our previous works, raw reads were generated by the vendor CosmosID Inc. (Germantown, MD, USA) using the fecal pellets from both LEAN and LEAN+CYN mice groups [36]. In brief, fecal pellets from mice were used for the isolation and purification of total DNA using the ZymoBIOMICS Miniprep kit. DNA library was prepared using the NexteraXT kit. HiSeq X platform was used for whole-genome sequencing and vendor-optimized protocol was used. For sequencing, an average insert size of 1350 bp and 2 × 150 bp of read length was used. After obtaining the data, the raw data were stored in Amazon AWS and run through fastqc after which a multiqc report was generated. Further, they were checked to ensure that there was no abnormality with duplication rates, adaptor content, or read quality and it was ensured that the read depth thresholds were met. Finally, the taxonomic results were reviewed on http://app.cosmosid.com (accessed on 15 April 2022) to confirm that there was no issue regarding barcoding or contamination.

### 4.4. Laboratory Methods

#### 4.4.1. Histopathology

Formalin-Fixed, paraffin-embedded, 5 μm-thick liver sections were stained with hematoxylin and eosin (H&E) at AML Laboratories (St. Augustine, FL, USA). Picrosirius red staining was performed at the Instrument Resources Facility, University of South Carolina School of Medicine (Columbia, SC, USA) to detect collagen fiber deposition in the liver. Both H&E and Picrosirius red-stained liver sections were observed under an Olympus BX43 microscope (Olympus America, Center Valley, PA, USA) using the 20× objective.

#### 4.4.2. Immunohistochemistry

The deparaffinization procedure of paraffin-embedded liver and small intestine slices was performed following our standard laboratory procedure. All 5 μm-thick tissue sections were immersed in 100% xylene first, followed by a 1:1 solution of xylene and ethanol, then 100% ethanol, 95% ethanol, 70% ethanol, and 50% ethanol in succession, and finally in deionized water for 3 min each. After the deparaffinization procedure, antigen epitope retrieval was done using the epitope retrieval solution and steamer (IHC-World, Woodstock, MD, USA). Blocking of endogenous peroxidase activity was carried out using a 3% H_2_O_2_ solution for 20 min, followed by serum blocking with 5% goat serum for 1 h. Once serum blocking was completed, anti-HMGB1, anti-CD68, anti-α-SMA, anti-IL-1β, and anti-TGF-β primary antibodies were diluted (1:300 dilution) in the same blocking buffer and applied onto the tissues. All the tissue sections were kept at 4 °C for overnight incubation in a humidified chamber. Following the overnight incubation period, all tissue sections were washed with 1X PBS-T (PBS+ 0.05% Tween 20) 3 times. Biotinylated secondary antibodies (species-specific) were probed to the tissue sections (1:250 dilution), followed by incubation with Strp-HRP (1:200 dilution). Lastly, the chromogenic substrate solution of 3,3′-Diaminobenzidine (DAB) (Abcam, Cambridge, MA, USA) was applied to the sections, whereas Mayer’s hematoxylin (Sigma-Aldrich, St. Louis, MO, USA) was used for performing counterstaining. All the liver and small intestine tissue sections were mounted with Simpo mount (GBI Laboratories, Mukilteo, WA, USA). The reactivity of the applied antibodies in the sections was observed under the 20× objective. All the immunohistochemistry images for this study were captured using an Olympus BX43 microscope (Olympus America, Center Valley, PA, USA) and morphometric data of the images were analyzed using CellSens Software from Olympus America (Center Valley, PA, USA).

#### 4.4.3. Immunofluorescence Staining and Microscopy

Deparaffinization and epitope retrieval procedures of the paraffin-embedded liver and small intestine sections were carried out in an exactly similar manner as mentioned for the immunohistochemistry method. After the epitope retrieval process was completed, all tissue sections were permeabilized using PBS-Tx (PBS+ 0.1% Triton X-100) solution for 1 h, followed by blocking with 5% goat serum for 1 h. Following the blocking step, the tissue sections were probed with anti-Claudin2, anti-Occludin, anti-NLRP3, anti-ASC2, anti-Smad2/3, and anti-Smad4 primary antibodies (1:300 dilution) and kept overnight at 4 °C in a humidified chamber. After that, species-specific anti-IgG secondary antibodies conjugated with Alexa Fluor 488 or 633 from Invitrogen (Rockford, IL, USA) were applied onto the sections (1:250 dilution). Lastly, ProLong Gold antifade reagent with DAPI (Life Technologies, Carlsbad, CA, USA) was used to mount the tissue sections. All immunofluorescence-stained images for this study were captured by an Olympus BX63 microscope (Olympus America, Center Valley, PA, USA) using the 40× objective. Analyses of all morphometric data were performed using CellSens Software from Olympus America (Center Valley, PA, USA).

#### 4.4.4. Quantitative Real-Time Polymerase Chain Reaction (qRT-PCR)

Levels of gene expression in the liver tissue samples were measured by the two-step qRT-PCR protocol. Firstly, all liver tissue samples obtained from each mouse were homogenized in TRIzol reagent (Invitrogen, Rockford, IL, USA) and then centrifuged to eliminate any excess tissue particles and debris. Following homogenization and centrifugation procedures, total RNA from the individual liver sample was isolated and purified using RNAse mini kit columns (Qiagen, Valencia, CA, USA) as per the manufacturer’s protocol. Then, purified RNA (1000 ng) was converted to cDNA using the iScript cDNA synthesis kit (Bio-Rad, Hercules, CA, USA) following the manufacturer’s standard procedure. Finally, qRT-PCR was carried out with the gene-specific mouse primers using SsoAdvanced SYBR Green supermix (Bio-Rad, Hercules, CA, USA) and CFX96 thermal cycler (Bio-Rad, Hercules, CA, USA). Threshold cycle (C_t_) values for the selected genes were normalized against 18S (internal control) values in the same sample. The relative fold change was calculated by the 2^−ΔΔCt^ method using the LEAN group of mice as the control. The sequences (5′-3′ orientation) for the mouse-specific primers used for real-time PCR are mentioned in Table 1.

#### 4.4.5. Western Blot

Proteins were extracted from liver samples using 1× RIPA lysis buffer containing protease and phosphatase inhibitors. The extracted tissue protein concentration was determined using the BCA assay kit (Thermo Fisher Scientific, Waltham, MA, USA). Approximately 40 µg of extracted protein from the individual liver sample was mixed with 1× NuPAGE™ LDS Sample Buffer (Thermo Fisher Scientific, Waltham, MA, USA) and 10% β-mercaptoethanol, followed by boiling for 5 min. After that, the extracted protein samples from the liver were subjected to standard SDS-PAGE using Novex 4–12% bis-tris gradient gel. The process of membrane transfer of the separated protein bands on a nitrocellulose membrane was carried out using the Trans-Blot Turbo transfer system (Bio-Rad, Hercules, CA, USA), and the membrane was blocked with 3% bovine serum albumin (BSA) for 1 h. Anti-BAX, anti-Bcl-2, anti-Cleaved PARP, anti-PARP, anti-Cleaved Caspase 9, anti-Caspase 9, anti-Cleaved Caspase 3, anti-Caspase 3, anti-HMGB1, anti-TLR4, anti-RAGE, anti-MyD88, and anti-β-actin primary antibodies were probed (1:1000 dilution) and kept at 4 °C overnight. Following 3 washes with 1X TBS-T (Tris-buffered saline+0.05% Tween 20) buffer, compatible species-specific HRP-conjugated secondary antibodies were applied (1:2500 dilution). Pierce ECL Western blotting substrate (Thermo Fisher Scientific, Waltham, MA, USA) was used to develop the blots. Finally, the images of blots were captured by G: Box Chemi XX6. Image J software (NIH, Bethesda, MD, USA) was used for all densitometric analyses.

#### 4.4.6. Enzyme-Linked Immunosorbent Assay (ELISA)

Serum HMGB1 level (ng/mL) in the LEAN and LEAN+CYN groups of mice was quantified using the ELISA kit from Abclonal Technology (Woburn, MA, USA) following the manufacturer’s standard protocol. Firstly, standard and serum samples were diluted using the supplied sample diluent and then applied to the designated wells for incubation. Following this, washing was performed, and the biotin-conjugated antibody solution was applied to each well. This was followed by another washing process and the addition of Streptavidin-HRP solution. After a subsequent washing process, TMB (3,3′,5,5′-Tetramethylbenzidine) solution was added to each well for color development, and the reaction was stopped by adding the stop solution. The required optical density value for each well was determined using a microplate reader. The HMGB1 level for each serum sample was calculated using the standard curve.

#### 4.4.7. Endotoxemia Detection by Limulus Amebocyte Lysate (LAL) Assay

Serum endotoxin concentration (EU/mL) was quantified in the LEAN and LEAN+CYN mice groups using the Pierce LAL Chromogenic Endotoxin Quantitation Kit from Thermo Fisher Scientific (Waltham, MA, USA) following the manufacturer’s standard protocol. First, the serum samples were diluted, and reagents and standards were prepared as per the instructions. Then, the assay plate was pre-equilibrated at 37 °C and the temperature was maintained throughout the assay procedure. Diluted serum samples and standard solutions were applied to the wells followed by reconstituted Amebocyte Lysate Reagent. Subsequently, the reconstituted chromogenic substrate solution was added to the wells and incubated for 6 min. Then, a 25% acetic acid solution was used as the stop solution, and the required optical density value was measured for each well using a microplate reader. The endotoxin level for each serum sample was calculated using the standard curve.

### 4.5. Statistical Analyses

All statistical analyses for this study were performed using GraphPad Prism software version 9.4.1. (San Diego, CA, USA). All data for this study are presented as the mean ± SEM. Unpaired *t*-tests (two-tailed tests with equal variance) were carried out to determine inter-group comparison, followed by Bonferroni–Dunn post hoc corrections analysis. For all analyses, *p* ≤ 0.05 was considered statistically significant.

## Figures and Tables

**Figure 1 toxins-14-00835-f001:**
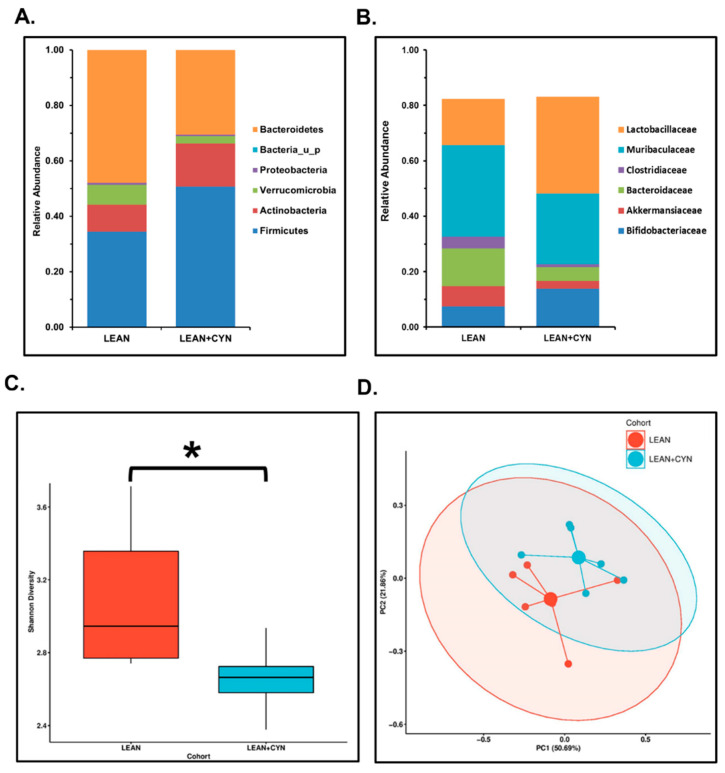
**CYN exposure in mice for 15 days via oral gavage caused an altered gut bacteriome pattern at both phylum and family levels with decreased α-diversity.** (**A**) The relative abundance of the gut bacteriome at the phylum level for LEAN (mice treated with vehicle only) and LEAN+CYN (mice administered CYN for 15 days via oral gavage) groups is presented by group average. (**B**) The relative abundance of the gut bacteriome at the family level for LEAN and LEAN+CYN groups is presented by group average. (**C**) Box plot depicting α-diversity (Shannon diversity) in both experimental groups (* *p* < 0.05). (**D**) Bray–Curtis β-diversity plot in both the LEAN and LEAN+CYN groups.

**Figure 2 toxins-14-00835-f002:**
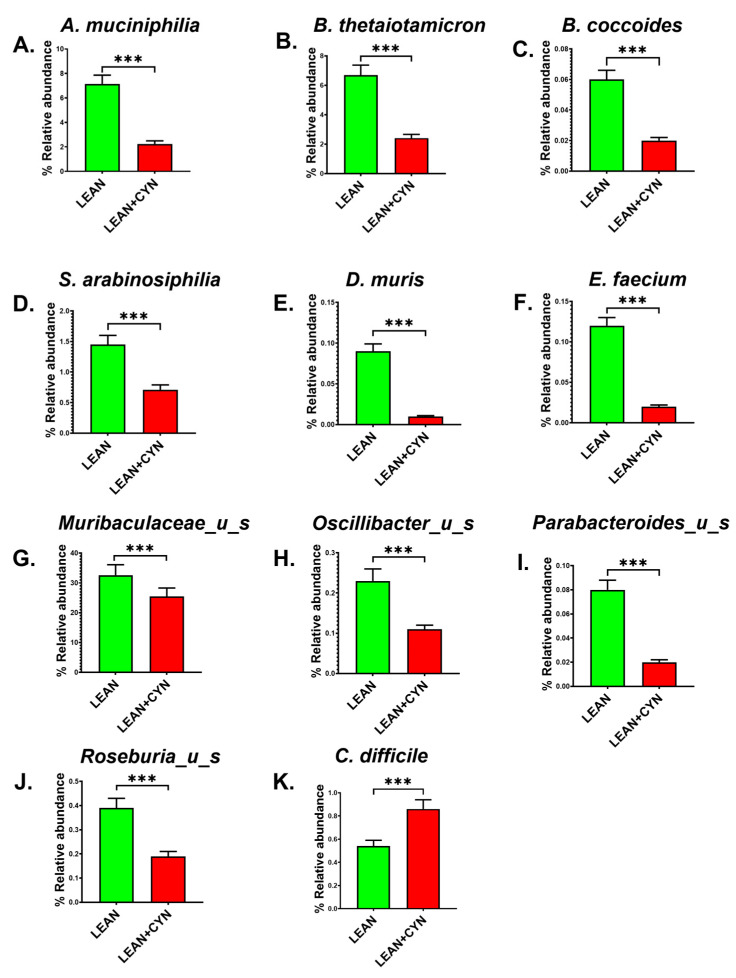
**CYN exposure in mice for 15 days via oral gavage caused an altered gut bacteriome pattern at the species level.** Bar graphs showing the percentage relative abundance of eleven significantly altered bacteria, (**A**) *Akkermansia muciniphila*, (**B**) *Bacteroides thetaiotaomicron*, (**C**) *Blautia coccoides*, (**D**) *Schaedlerella arabinosiphila*, (**E**) *Duncaniella muris*, (**F**) *Enterococcus faecium*, (**G**) *Muribaculaceae_u_s*, (**H**) *Oscillibacter_u_s*, (**I**) *Parabacteroides_u_s*, (**J**) *Roseburia_u_s*, and (**K**) *Clostridioides difficile* at the species level between the LEAN and LEAN+CYN groups (*** *p* < 0.001). Data were represented as mean ± SEM and statistical significance was tested using unpaired *t*-test between the two groups (*** *p* < 0.001), followed by Bonferroni–Dunn post hoc corrections.

**Figure 3 toxins-14-00835-f003:**
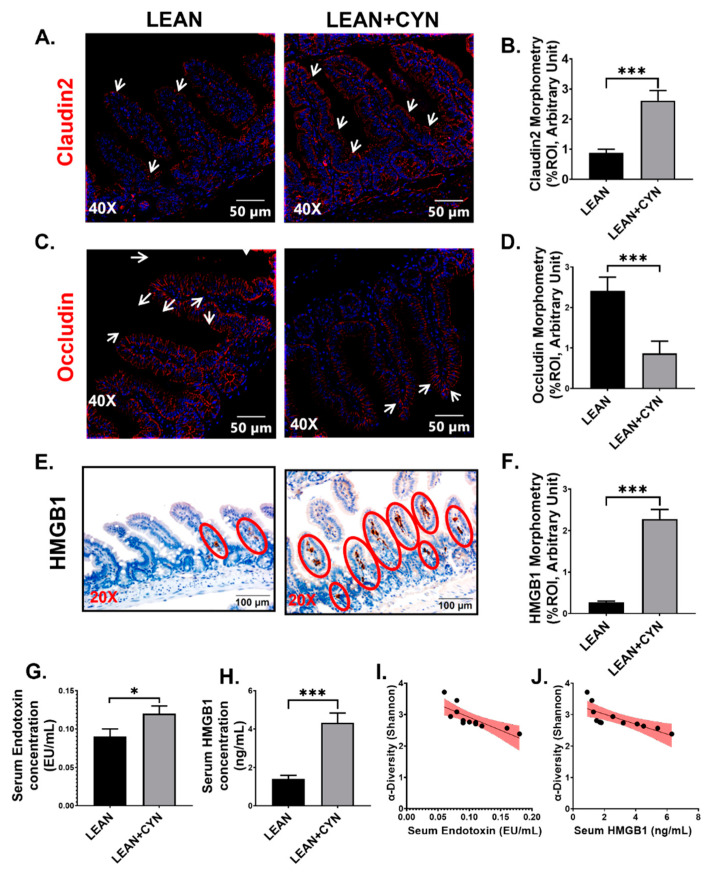
**CYN administration in mice results in gut leaching, increased gut HMGB1 expression, and elevated systemic endotoxin and HMGB1 levels.** Representative immunofluorescence images of (**A**) Claudin2 and (**C**) Occludin immunoreactivity (red) and (**E**) immunohistochemistry images of HMGB1 immunoreactivity in the small intestine sections of LEAN, and LEAN+CYN mice groups. For immunofluorescence images, the small intestine sections were counterstained with DAPI (blue), all the images were captured in 40× magnification, and immunoreactivity was indicated by white arrows. For immunohistochemistry, all the images were captured in 20× magnification, and immunoreactivity was indicated by red circles. Morphometric analysis (calculated as %ROI) of (**B**) Claudin2, (**D**) Occludin, and (**F**) HMGB1 immunoreactivity where Y-axis represents % positive immunoreactive area (% ROI) (*n* = 3; mean value taken from three separate microscopic fields) (*** *p* < 0.001). The concentration of (**G**) Endotoxin (EU/mL) and (**H**) HMGB1 (ng/mL) levels were measured in the serum of both LEAN and LEAN+CYN mice groups and represented as bar graphs (* *p* < 0.05, *** *p* < 0.001). Data were represented as mean ± SEM and statistical significance was tested using unpaired *t*-test between the two groups (* *p* < 0.05, *** *p* < 0.001), followed by Bonferroni Dunn Post hoc corrections. Correlation plot between α-diversity index (Shannon Diversity) of the gut microbiome and serum (**I**) Endotoxin, and (**J**) HMGB1 concentrations. Pearson’s linear regression is represented in red with 95% confidence bands.

**Figure 4 toxins-14-00835-f004:**
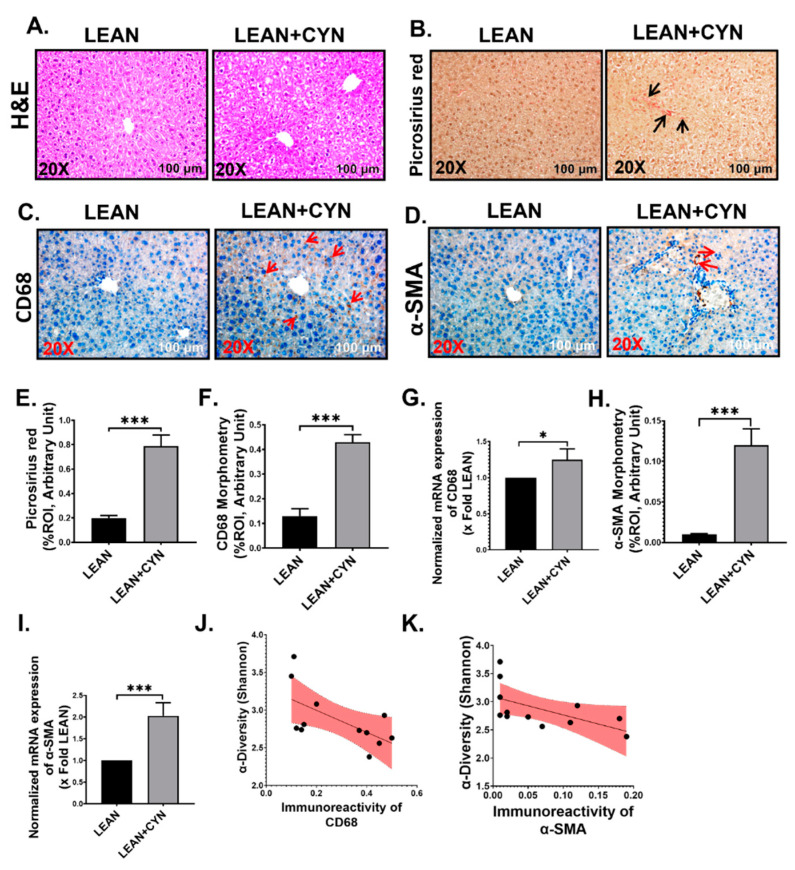
**CYN administration in mice led to liver injury, mild fibrosis, increased mRNA and protein level expression of Kupffer cells, and hepatic stellate cell activation biomarkers.** (**A**) Hematoxylin and eosin (H&E) staining indicated signs of liver injury in the LEAN+CYN mice group compared to the LEAN mice group. (**B**) Picrosirius red staining showed mild fibrotic pathology (indicated by black arrows) in the liver sections of the LEAN+CYN group compared to the LEAN control. Representative immunohistochemistry images of (**C**) CD68 and (**D**) α-SMA immunoreactivity was shown in the liver sections of LEAN, and LEAN+CYN groups. The images were captured in 20× magnification and the immunoreactivity of CD68 and α-SMA was indicated by red arrows. (**E**) Morphometric analysis of Picrosirius red staining estimated a significantly increased deposition of collagen fiber in the LEAN+CYN group compared to the LEAN group. (*** *p* < 0.001). Y-axis represents percentages of fibrosis expressed as %ROI. Morphometric analysis (calculated as %ROI) of (**F**) CD68 and (**H**) α-SMA immunoreactivity where Y-axis represents % positive immunoreactive area (% ROI) (*n* = 3; mean value taken from three separate microscopic fields) (*** *p* < 0.001). Normalized mRNA expression of (**G**) CD68 and (**I**) α-SMA against 18S in the livers of LEAN and LEAN+CYN groups of mice shown as fold change of the LEAN group (* *p* < 0.05, *** *p* < 0.001). Data were represented as mean ± SEM and statistical significance was tested using unpaired *t*-test between the two groups (* *p* < 0.05, *** *p* < 0.001), followed by Bonferroni–Dunn post hoc corrections. Correlation plot between α-diversity index (Shannon diversity) of the gut microbiome and (**J**) CD68, and (**K**) α-SMA immunoreactivity in the liver. Pearson’s linear regression is represented in red with 95% confidence bands.

**Figure 5 toxins-14-00835-f005:**
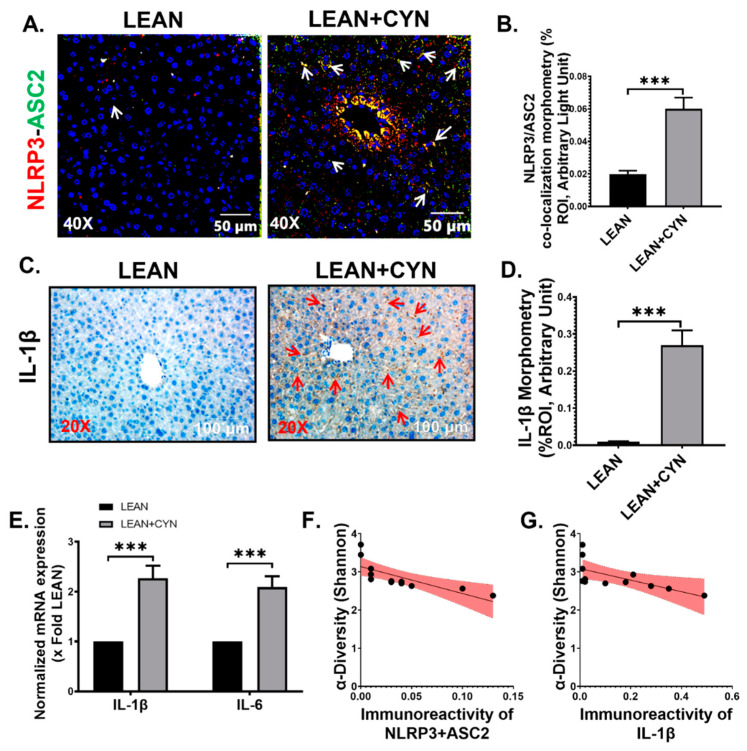
**CYN exposure in mice caused NLRP3 inflammasome activation and increased mRNA and protein level expression of pro-inflammatory cytokines in the liver.** Representative (**A**) immunofluorescence images showing NLRP3 (red) and ASC2 (green) co-localization events and (**C**) immunohistochemistry images of IL-1β immunoreactivity in the liver sections of LEAN and LEAN+CYN mice groups. For immunofluorescence images, the liver sections were counterstained with DAPI (blue); all the images were captured in 40× magnification, and immunoreactivity was indicated by white arrows. For immunohistochemistry, all the images were captured in 20× magnification, and immunoreactivity was indicated by red arrows. Morphometric analysis (calculated as %ROI) of (**B**) NLRP3 and ASC2 co-localization events, and (**D**) IL-1β immunoreactivity where *Y*-axis represents % positive immunoreactive area (% ROI) (*n* = 3; mean value taken from three separate microscopic fields) (*** *p* < 0.001). (**E**) Normalized mRNA expression of IL-1β and IL-6 against 18S in the livers of LEAN and LEAN+CYN groups of mice and shown as fold change of the LEAN group (*** *p* < 0.001). Data were represented as mean ± SEM and statistical significance was tested using unpaired *t*-test between the two groups (*** *p* < 0.001), followed by Bonferroni–Dunn post hoc corrections. Correlation plot between α-diversity index (Shannon diversity) of the gut microbiome and (**F**) NLRP3-ASC2 co-localization events, and (**G**) IL-1β immunoreactivity in the liver. Pearson’s linear regression is represented in red with 95% confidence bands.

**Figure 6 toxins-14-00835-f006:**
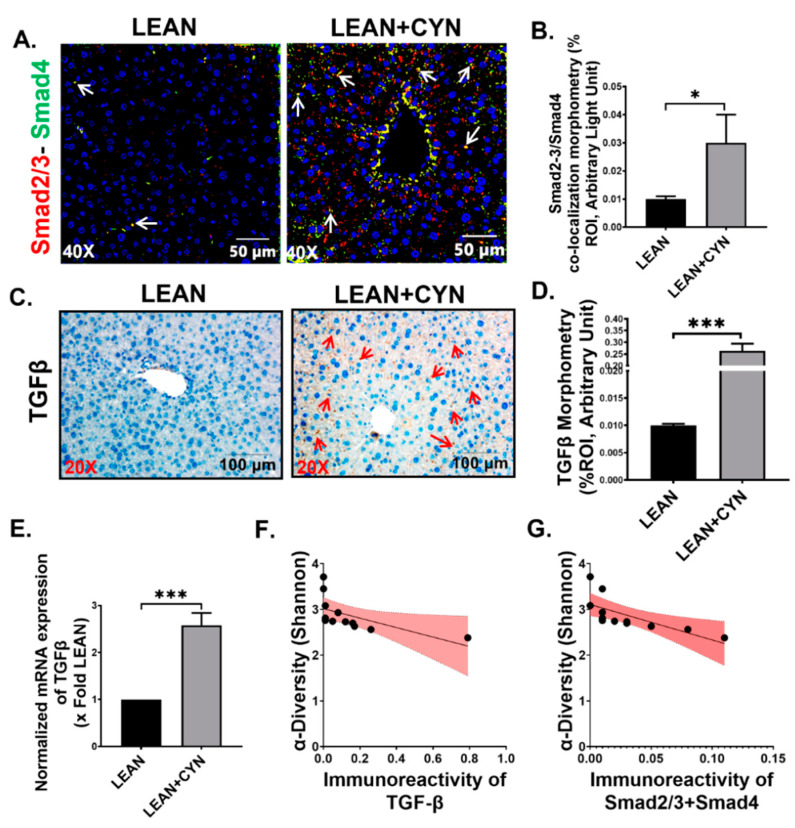
**CYN administration in mice activates TGF-β mediated Smad2/3-Smad4 fibrotic pathway in the liver.** Representative (**A**) immunofluorescence images showing Smad2/3 (red) and Smad4 (green) co-localization events and (**C**) immunohistochemistry images of TGF-β immunoreactivity in the liver sections of LEAN, and LEAN+CYN mice groups. For immunofluorescence images, the liver sections were counterstained with DAPI (blue), all the images were captured in 40× magnification, and immunoreactivity was indicated by white arrows. For immunohistochemistry, all the images were captured in 20× magnification, and immunoreactivity was indicated by red arrows. Morphometric analysis (calculated as %ROI) of (**B**) Smad2/3 and Smad4 co-localization events, and (**D**) TGF-β immunoreactivity where Y-axis represents % positive immunoreactive area (% ROI) (*n* = 3; mean value taken from three separate microscopic fields) (*** *p* < 0.001, * *p* < 0.05). (**E**) Normalized mRNA expression of TGF-β against 18S in the livers of LEAN and LEAN+CYN group and shown as fold change of the LEAN group (*** *p* < 0.001). Data were represented as mean ± SEM and statistical significance was tested using unpaired *t*-test between the two groups (* *p* < 0.05, *** *p* < 0.001), followed by Bonferroni Dunn Post hoc corrections. Correlation plot between α-diversity index (Shannon Diversity) of the gut microbiome and (**F**) TGF-β immunoreactivity, and (**G**) NLRP3-ASC2 co-localization events in the liver. Pearson’s linear regression is represented in red with 95% confidence bands.

**Figure 7 toxins-14-00835-f007:**
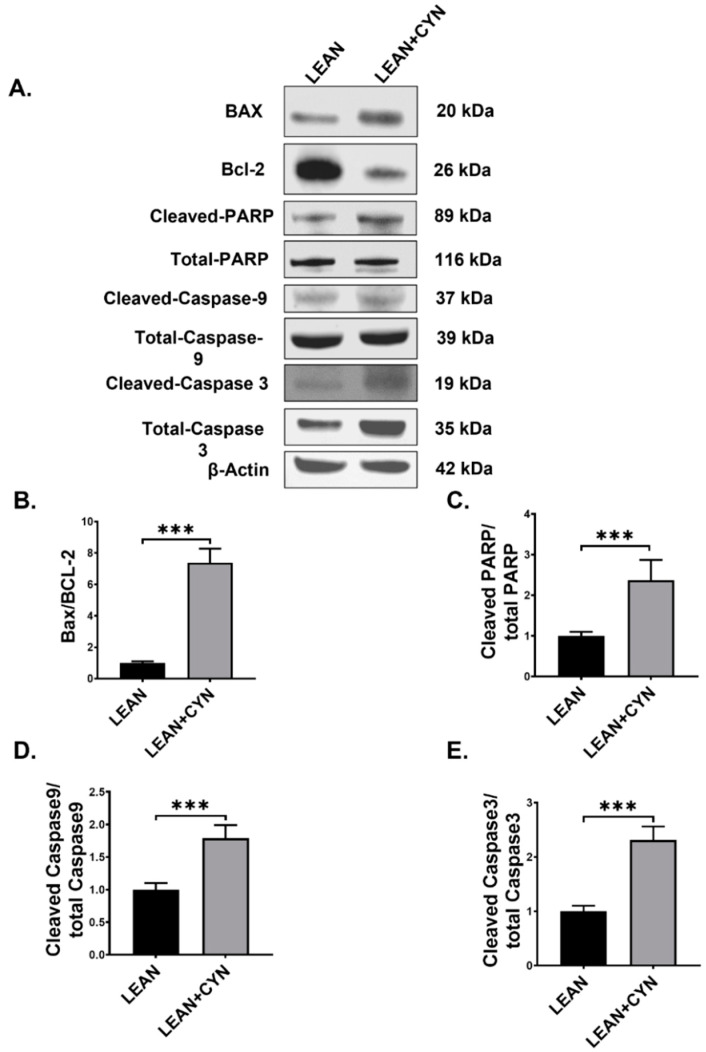
**CYN exposure in mice induced the intrinsic apoptotic pathway in the liver.** (**A**) Western blot images of BAX, Bcl-2, Cleaved-PARP, Total-PARP, Cleaved-Caspase 9, Total-Caspase 9, Cleaved-Caspase 3, Total-Caspase 3, and β-actin protein expression levels were obtained from liver tissue extracts from the LEAN and LEAN+CYN groups of mice. Densitometry analyses of (**B**) BAX protein expression normalized against Bcl-2 expression, (**C**) Cleaved-PARP protein expression normalized against Total-PARP expression, (**D**) Cleaved-Caspase 9 protein expression normalized against Total-Caspase 9 expression, and (**E**) Cleaved-Caspase 3 protein expression normalized against Total-Caspase 3 expression (*** *p* < 0.001) and plotted as bar graphs. Data were represented as mean ± SEM and statistical significance was tested using unpaired *t*-test between the two groups (*** *p* < 0.001), followed by Bonferroni– Dunn post hoc corrections.

**Figure 8 toxins-14-00835-f008:**
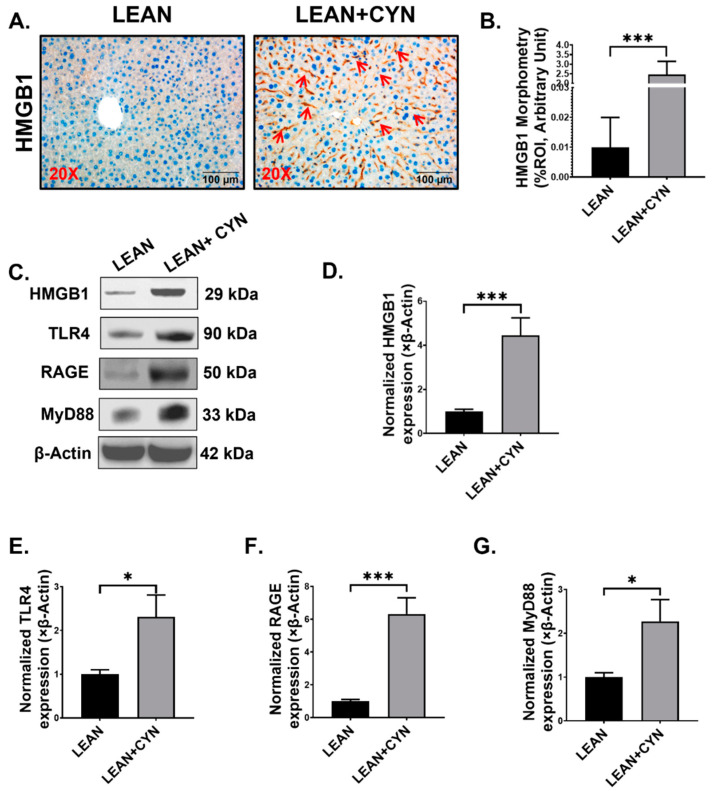
**CYN exposure in mice caused HMGB1 secretion and increased expression of its receptors.** (**A**) Representative immunohistochemistry images of HMGB1 immunoreactivity shown in the LEAN and LEAN+CYN groups. The images were captured in 20× magnification and the immunoreactivity of HMGB1 was indicated by red arrows. (**B**) Morphometric analysis (calculated as %ROI) of HMGB1 immunoreactivity where Y-axis represents % positive immunoreactive area (% ROI) (*n* = 3; mean value taken from three separate microscopic fields) (*** *p* < 0.001). (**C**) Western blot images of HMGB1, TLR4, RAGE, MyD88, and β-actin protein expression levels were obtained from liver tissue extracts from the LEAN and LEAN+CYN groups of mice. Densitometry analyses of (**D**) HMGB1, (**E**) TLR4, (**F**) RAGE, and (**G**) MyD88 protein expressions normalized against β-actin protein expression (* *p* < 0.05, *** *p* < 0.001) and plotted as bar graphs. Data were represented as mean ± SEM and statistical significance was tested using unpaired *t*-test between the two groups (* *p* < 0.05, *** *p* < 0.001), followed by Bonferroni–Dunn post hoc corrections.

**Figure 9 toxins-14-00835-f009:**
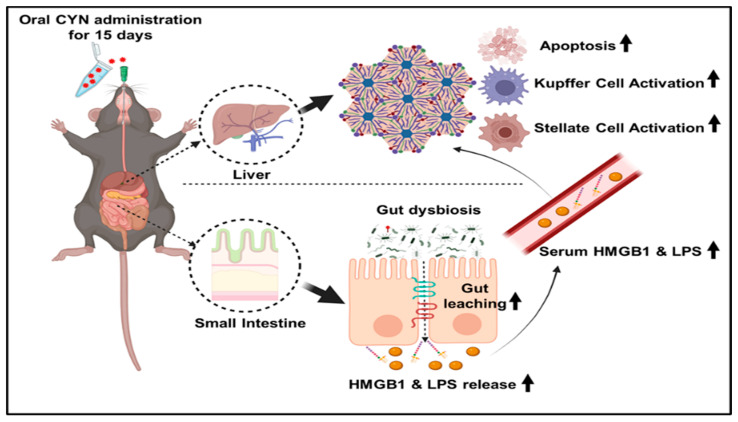
Graphical abstract representing the effects of CYN administration in mice for a continuous period of 15 days leading to gut dysbiosis, increased intestinal permeability, elevated amount of serum endotoxin and HMGB1, and multiple pathophysiological symptoms in the liver.

**Table 1 toxins-14-00835-t001:** qRT-PCR primer sequence.

Genes	Primer Sequence (5′-3′ Orientation)
CD68 (*Mus musculus*)	Forward: GCTACATGGCGGTGGAGTACAA Reverse: ATGATGAGAGGCAGCAAGATGG
α-SMA (*Mus musculus*)	Forward: GGAGAAGCCCAGCCAGTCGC Reverse: ACCATTGTCGCACACCAGGGC
IL-1β (*Mus musculus*)	Forward: CCTCGGCCAAGACAGGTCGC Reverse: TGCCCATCAGAGGCAAGGAGGA
IL-6 (*Mus musculus*)	Forward: ACCAGAGGAAATTTTCAATAGGC Reverse: TGATGCACTTGCAGAAAACA
TGF-β (*Mus musculus*)	Forward: CTCACCGCGACTCCTGCTGC Reverse: TCGGAGAGCGGGAACCCTCG
18S (*Mus musculus*)	Forward: TTCGAACGTCTGCCCTATCAA Reverse: ATGGTAGGCACGGCGATA

## Data Availability

The data presented in this study are available on request from the corresponding author. Microbiome sequence data of this study have been deposited in GenBank.

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
