# Peer review of "Subchronic Oral Cylindrospermopsin Exposure Alters the Host Gut Microbiome and Is Associated with Progressive Hepatic Inflammation, Stellate Cell Activation, and Mild Fibrosis in a Preclinical Study"

_toxins, 2022, doi:10.3390/toxins14120835_

Round 1

Reviewer 1 Report

Thank you for allowing me to review this interesting article that deals with the influence of CYN exposure on mice's gut microbiome and how its alteration can lead to different effects at the hepatic level. It is a well-designed study and a well-written article, which needs some revisions to be published in Toxins.

First, the theoretical framework is not fully framed. Although the origin of cyanotoxins and their possible toxic mechanisms are perfectly described, the introduction should also establish the theoretical framework that relates their effect on the gut microbiota and how this alteration can influence the liver.

On the other hand, in the discussion, between lines 396 and 400, I miss one or two citations supporting the authors' assertion.

In the methodology, I have a question: could exposure to anaesthesia alter the research results? I am not used to working with animals, but most drugs tend to have a hepatotoxic effect. How could this affect the gut microbiota?

Finally, I suggest that the authors expand on points 4.4.6 and 4.4.7. Although these protocols are carried out following the protocol supplied by the manufacturer, a brief description would be appreciated. Likewise, it is recommended that point 4.5 be described in a little more detail.

I am convinced that once these aspects have been revised in the original manuscript, it will be ready for publication.

Author Response

Reviewer 1. Comments and Suggestions for Authors

Thank you for allowing me to review this interesting article that deals with the influence of CYN exposure on mice's gut microbiome and how its alteration can lead to different effects at the hepatic level. It is a well-designed study and a well-written article, which needs some revisions to be published in Toxins.

Response: First, we want to thank the reviewer for applauding our work and providing us with insightful input.

First, the theoretical framework is not fully framed. Although the origin of cyanotoxins and their possible toxic mechanisms are perfectly described, the introduction should also establish the theoretical framework that relates their effect on the gut microbiota and how this alteration can influence the liver.

Response: We want to thank the reviewer for this valuable suggestion. We have mentioned the gut microbiota and how it’s connected to the liver via the gut-liver axis in the introduction section briefly.

On the other hand, in the discussion, between lines 396 and 400, I miss one or two citations supporting the authors' assertion.

Response: We want to thank the reviewer for pointing out this ommission. We have added the citations in the revised form of our manuscript.

In the methodology, I have a question: could exposure to anaesthesia alter the research results? I am not used to working with animals, but most drugs tend to have a hepatotoxic effect. How could this affect the gut microbiota?

Response: We feel that anesthesia by itself can’t possibly change the results as the time of anesthesia/euthanasia process was too short for the animals for having an impact on the gut microbiome and was used before only the sacrifice process. If anesthesia would have any effect on the animals, we would also observe similar changes in the CONTROL group of mice regarding hepatotoxicity. Also, gut microbiome alteration is known to be not a transient process as the microbiome is relatively stable in the hosts, thus we are convinced that alteration of the gut microbiome due to a short exposure to anesthesia would be very unlikely.

Finally, I suggest that the authors expand on points 4.4.6 and 4.4.7. Although these protocols are carried out following the protocol supplied by the manufacturer, a brief description would be appreciated. Likewise, it is recommended that point 4.5 be described in a little more detail.

Response: We want to thank the reviewer for this valuable suggestion. We have included a brief description for both HMGB1 ELISA and serum endotoxin assay in the revised form of the manuscript as per the reviewer’s opinion.

I am convinced that once these aspects have been revised in the original manuscript, it will be ready for publication.

Reviewer 2 Report

The paper entitled: Subchronic oral cylindrospermopsin exposure alters the host gut microbiome and is associated with progressive hepatic inflammation, stellate cell activation, and mild fibrosis in a preclinical study, deals with the possible role of cylindrospermopsin, an emerging threat from cyanotoxins, in causing host gut dysbiosis and its association with liver pathology in a mouse model of toxico-pharmacokinetics and hepatic pathology, founding that oral exposure to CYN in mice caused decreased diversity of gut bacteria phyla accompanied by an increased abundance of C. difficile and decreased abundance of probiotic flora and progressive liver damage, providing some evidence of CYN-linked progressive liver pathology linked to gut dysbiosis. The experimental work is explained thoroughly and the conclusions are well founded although the could require further experimental work to take final conclusions. The report constitutes an incremental advance in the field and therefore can be published as it is. The paper is very long and probably some experimental part could be left in a supporting information section.

Author Response

The paper entitled: Subchronic oral cylindrospermopsin exposure alters the host gut microbiome and is associated with progressive hepatic inflammation, stellate cell activation, and mild fibrosis in a preclinical study, deals with the possible role of cylindrospermopsin, an emerging threat from cyanotoxins, in causing host gut dysbiosis and its association with liver pathology in a mouse model of toxico-pharmacokinetics and hepatic pathology, founding that oral exposure to CYN in mice caused decreased diversity of gut bacteria phyla accompanied by an increased abundance of C. difficile and decreased abundance of probiotic flora and progressive liver damage, providing some evidence of CYN-linked progressive liver pathology linked to gut dysbiosis. The experimental work is explained thoroughly and the conclusions are well founded although the could require further experimental work to take final conclusions. The report constitutes an incremental advance in the field and therefore can be published as it is. The paper is very long and probably some experimental part could be left in a supporting information section.

Response: We want to convey our gratitude to the reviewer for appreciating and accepting our research work. However, we feel that each experiment conducted in this study is highly essential for establishing the proposed hypothesis. Also, we will surely carry out further experiments in future studies based on our research findings in the current study. In addition, the experiments in this study have been conducted according to the ARRIVE guidelines, and we believe that it will be very helpful to the peers if we provide detailed information in the main manuscript rather than as a supporting section.

Reviewer 3 Report

The MS describes examined the possible role of cylindrospermopsin in causing host gut dysbiosis and its association with liver pathology in a mouse model of toxico-pharmacokinetics and hepatic pathology. The MS provides plenty of evidence of CYN-linked progressive liver pathology that may be well-linked to gut dysbiosis. I believe that the MS may contributions to the mechanism of influence of the gut dysbiosis. However, there are still several issues that need to be modified .

1. The title is somewhat too general, so a more definited title is suggested.

2. Abstrect, some irregular words should be cottrected. Such as Line 14, CYN; Line 15, C. difficile; Line 16, B. thetaiotamicron; please check.

3. Keywords, gut microbiome;

4. The language of the MS should be improved.

5. In section Results, I think it would be possible to simplify the content and highlight the point.

6. Check some sentenses in the MS, for example, "environmental exposure too also greatly influences the host microbiome" in Line 383.

7. As noted in the abstract, changes in the diversity of gut microbiome are often associated with intestinal and liver pathology and potential gastrointestinal diseases. It is suggested to add relevant research data in the introduction to make the introduction of research background more clearer.

8. Hepatotoxicity mechanism of cylindrospermopsin was proposed in the MS, and the experimental data seemed to relatively complete, but the synergistic data between the diversity of gut dysbiosis and the hepatotoxicity of the cylindrospermopsin were insufficient. It is suggested to add more discussion on synergistic data to strengthen the association between the gut dysbiosis and hepatotoxicity.

9. References: please correct the format. and the number of the references is suggested to reduce. 80 references are too much for an article.

Author Response

Reviewer 3. Comments and Suggestions for Authors

The MS describes examined the possible role of cylindrospermopsin in causing host gut dysbiosis and its association with liver pathology in a mouse model of toxico-pharmacokinetics and hepatic pathology. The MS provides plenty of evidence of CYN-linked progressive liver pathology that may be well-linked to gut dysbiosis. I believe that the MS may contributions to the mechanism of influence of the gut dysbiosis. However, there are still several issues that need to be modified .

  1. The title is somewhat too general, so a more definited title is suggested.

Response: The title contains all the key words that we feel justifies the salient points of the manuscript. We respectfully disagree with the comment.

  1. Abstrect, some irregular words should be cottrected. Such as Line 14, CYN; Line 15, C. difficile; Line 16, B. thetaiotamicron; please check.

Response: We want to thank the reviewer for detecting the mistake. We have corrected these issues in the revised version of the manuscript (Lines 14-16, Line 24).

  1. Keywords, gut microbiome;

Response: We want to thank the reviewer for the suggestion and we have changed the keyword to ‘gut microbiome’ (Line 27).

  1. The language of the MS should be improved.

Response: We want to thank the reviewer for the suggestion and we have made the necessary changes to improve the overall language in the manuscript.

  1. In section Results, I think it would be possible to simplify the content and highlight the point.

Response: We want to thank the reviewer for the suggestion and we have made the necessary changes in the manuscript.

  1. Check some sentenses in the MS, for example, "environmental exposure too also greatly influences the host microbiome" in Line 383.

Response: We want to thank the reviewer for detecting the mistake. We have corrected it in the revised version of the manuscript.

  1. As noted in the abstract, changes in the diversity of gut microbiome are often associated with intestinal and liver pathology and potential gastrointestinal diseases. It is suggested to add relevant research data in the introduction to make the introduction of research background more clearer.

Response: As per the reviewer’s suggestion, we have mentioned the role of the gut microbiome, and its connection to the liver in the introduction section to make the research background clearer (Lines: 83-100).

  1. Hepatotoxicity mechanism of cylindrospermopsin was proposed in the MS, and the experimental data seemed to relatively complete, but the synergistic data between the diversity of gut dysbiosis and the hepatotoxicity of the cylindrospermopsin were insufficient. It is suggested to add more discussion on synergistic data to strengthen the association between the gut dysbiosis and hepatotoxicity.

Response: We want to thank the reviewer for the suggestion. As per the reviewer’s instruction, we have mentioned the bacteriome diversity and its association with hepatotoxicity in the discussion section (Lines: 470-479).

  1. References: please correct the format. and the number of the references is suggested to reduce. 80 references are too much for an article.

Response: We want to thank the reviewer for the suggestion. However, we have this practice in our laboratory to preferably cite the original research articles in our manuscripts rather than review articles. This might have caused the increased number of references as we had to cite a lot of original research articles in the manuscript. However, as per the suggestion made by the reviewer, we will definitely try to reduce the number of references in the future.

Reviewer 4 Report

The problems caused by toxic cyanobacterial blooms around the world have always provoked the attention of scientists. Moreover, in recent years, they have become more frequent. The effects that cyanotoxins (hepato-, neuro- and dermatotoxins) have as a risk to the environment and human health are the subject of many research groups. Microcystins have been very well studied as the most common hepatotoxins, and less attention has been paid to other cyanotoxins, including CYN. In this regard, I find the present study relevant and significant. Additionally, the relationship between CYN and the microbiome is being explored, which is extremely interesting in itself. The study design is well done. Results obtained are interesting and significant, and excellently presented. Research methods are modern, varied and very well selected. I have no objections to the work of the author team and I express my admiration for this type of research.

Author Response

The problems caused by toxic cyanobacterial blooms around the world have always provoked the attention of scientists. Moreover, in recent years, they have become more frequent. The effects that cyanotoxins (hepato-, neuro- and dermatotoxins) have as a risk to the environment and human health are the subject of many research groups. Microcystins have been very well studied as the most common hepatotoxins, and less attention has been paid to other cyanotoxins, including CYN. In this regard, I find the present study relevant and significant. Additionally, the relationship between CYN and the microbiome is being explored, which is extremely interesting in itself. The study design is well done. Results obtained are interesting and significant, and excellently presented. Research methods are modern, varied and very well selected. I have no objections to the work of the author team and I express my admiration for this type of research.

Response: We want to sincerely thank the reviewer for appreciating and accepting our current research work. These words of encouragement will surely motivate us to conduct in-depth, mechanistic studies in the future to further uncover various aspects of CYN toxicity.

Round 2

Reviewer 1 Report

I would like to congratulate the authors for their efforts. The changes made have improved the initial version of the manuscript. The article is now ready, from my point of view, to be published in Toxins.